# FEM and Analytical Modeling of the Incipient Chip Formation for the Generation of Micro-Features

**DOI:** 10.3390/ma14143789

**Published:** 2021-07-06

**Authors:** Michele Lanzetta, Armin Gharibi, Marco Picchi Scardaoni, Claudia Vivaldi

**Affiliations:** 1Research Center “E. Piaggio”, School of Engineering, University of Pisa, 56126 Pisa, Italy; 2Department of Civil and Industrial Engineering, University of Pisa, 56126 Pisa, Italy; marco.picchiscardaoni@ing.unipi.it (M.P.S.); armin.gharibi@phd.unipi.it (A.G.); claudia.95vivaldi@gmail.com (C.V.)

**Keywords:** micromachining, chip formation, finite element method, adhesion, sustainability

## Abstract

This paper explores the modeling of incipient cutting by Abaqus, LS-Dyna, and Ansys Finite Element Methods (FEMs), by comparing also experimentally the results on different material classes, including common aluminum and steel alloys and an acetal polymer. The target application is the sustainable manufacturing of gecko adhesives by micromachining a durable mold for injection molding. The challenges posed by the mold shape include undercuts and sharp tips, which can be machined by a special diamond blade, which enters the material, forms a chip, and exits. An analytical model to predict the shape of the incipient chip and of the formed grove as a function of the material properties and of the cutting parameters is provided. The main scientific merit of the current work is to approach theoretically, numerically, and experimentally the very early phase of the cutting tool penetration for new sustainable machining and micro-machining processes.

## 1. Introduction

The rise of miniaturization and micro-components signals the need for more research to be carried out on micro-machining, whose areas of applications include micro-injection molds, watch components, optical devices, and products for the aerospace, biomedical and electronic industries. This also prompts thinking about different machinable materials (metallic alloys, composites, polymers, and ceramics) and their behaviors relative to micromachining [1].

Although based on the same principle as *macro* cutting, the phenomena involved in *micro* cutting are not a simple scaling. A significant difference between these two cutting processes is the chip formation involving the ‘minimum chip thickness’ phenomenon.

Applications of micromachining include the development of surfaces with a microscopic geometry (or texturing) designed to achieve or enhance certain functional characteristics. Examples range from highly hydrophobic surfaces that reject liquids and dirt to directional dry adhesive materials that stick only when loaded in a particular direction. In many cases, these surfaces are inspired by examples found in nature, such as the lotus leaf, the skin of sharks, or the hierarchical adhesive apparatus found on the feet of geckos and certain arthropods. Reviews can be found in [2,3,4,5]. In most cases, the fabrication of these surfaces begins with lithographic patterning or direct micromachining of a material such as SU-8 [6], wax [7], or silicon to create a mold, which is used to cast the functional material. Alternative approaches are based on additive manufacturing, such as shape deposition manufacturing (SDM) [8] or direct laser lithography [9], but for large-scale manufacturing of these polymeric materials with functional surfaces, it is often desirable to employ a durable mold that can withstand elevated temperatures and pressures, and many cycles of molding and cleaning [10]. Many of the possible ways to create the desired mold geometry [11] have resulted in adhesives, which are lacking in some desirable combinations of properties as defined in [12,13,14] including very low preload, adhesion only when loaded in a preferred direction, low detachment effort, resistance to fouling, and a long lifetime. Some examples are displayed in Figure 1.

### Sustainable Manufacturing Processes

In the literature, there are many experimental approaches investigating the different ways of producing surfaces with a microscopic geometry [7,8,9,10,11,12,13,14,15,16,17,18,19]; the FEM has been used to model friction and adhesion of gecko materials in [15], but FEM analysis of the processes needs to be investigated and has not yet been.

This study presents the FEM modeling of incipient chip formation for the generation of micro features in two different material classes: metal and polymer.

The modeling of the incipient cutting process of these materials is based on the Johnson–Cook material model [13]. Although previous studies are a good starting point to model the chip formation process of the materials considered here, under various cutting parameters, further work is needed to predict the deformation response and the chip formation. Results and knowledge from experimental and numerical parts of this orthogonal cutting study can be utilized and interpreted for the micro machining process needed to fabricate micro geometries in aluminum and steel. Despite this being fundamental research, it has a direct impact on sustainability allowing a better understanding of the chip formation at the earliest stages, not yet systematically investigated in the literature. A faithful modeling of the interaction between tool and material allows the minimization of the energy required to trigger the chip formation.

## 2. Materials and Methods

The optimal cutting conditions to achieve the shape in Figure 1 were preliminarily investigated experimentally. Easy-to-machine materials have been selected, with a special form tool and cutting strategy shown in Figure 2.

Cutting was carried out on conventional 5-axis CNC machine (Deckel Maho DMU 60 Monoblock by DMG Mori, Nakamura-ku, Nagoya, Japan). The wedge-shaped tool was a poly-tetra-fluoroethylene (PTFE)-coated steel disposable microtome blade (Delaware Diamond Knives D554X Wilmington, DE, USA), with blade roughness on length scales ≪ 1 μm, an internal angle of β ≈ 24°, with a bevel that extends over the final 40 μm of the blade length and is 34° wide, and an edge radius < 0.9 μm. A special clamp (Figure 2, inset) was developed to ensure axial positioning with respect to the spindle axis of the CNC machine to allow pure rotation about the edge for cutting by two tools: a groove of few tens of microns needs to be machined again by two different tools.

Plastic extruded acetal co-polymer Delrin POM-C was selected due to its wide availability, machinability, and resistance, while it also minimized tool damage. The lubricant was favorable to prevent sticking of the long plastic chip, facilitated by the material thermal softening caused by the chip formation, occurring between 90 and 110° C at the glass transition.

For hard aluminum (Al 7075-T6) and steel (AISI 4140), achieved by alternative processes, such as machining by parallel passes as in Figure 3, limitations are evident, particularly on the excessive groove bottom radius, which should be avoided because the sharpness of radius is a fundamental requirement for the adhesion of the cast gecko material. Lower machinability and increased bottom radius are observed on the harder material (high performance steel). Macroscopic tool wear observed has further negatively affected the desired shape quality.

Figure 4a,b shows the improved quality of cutting by the blade according to the strategy in Figure 2. Figure 3 shows that the desired undercut geometry with small radius on the grove cannot be achieved in (a) because of the excessive angle β and in (b) because of the excessive radius. Favorable results were similarly observed for various blade inclinations (λ = 25, 27.5, 30, 32.5 and 35°). The blade was also tilted by the same angles along a trajectory of 30° (at the expenses of α and γ). The angle φ remains constant at 40° and at 60°, respectively.

The stressed material is confined to a prismatic region; the cavity depth d < 100 μm and its width is >10 mm, so it is reasonable to assume that the material is in a state of plane strain. This plane strain plasticity problem seems closer to orthogonal machining as opposed to oblique wedge indenting.

For the material conservation, the area of the protrusion is equivalent to the cut material and can be approximated to a triangle assuming the lost area due to curvature S2 ≈ S1; consequently, the protrusion height p and the angle η are bound by:p^2^ =C tan η(1)
where C is given by the cutting conditions:C = d^2^ (cot α − tan γ)(2)
which is applicable when γ > 0; is minimum for high α and is positive when α + γ < 90 (small β—thin cutters). C < 0 implies η < 0, which would be acceptable from a geometrical viewpoint, but would invalidate the hypothesis of material conservation. 

Considering the blade angle between cutters β and the trajectory λ
C = d^2^ [cot (λ − β/2) + cot (λ + β/2)] (3)

The direct fabrication of hard plastic molds allows manufacturing flexibility for small batches of adhesive patches, with negligible blade wear, and has been achieved as shown by the SEM micrograph in Figure 4b. Shaping by the insert produced deformation (top); the bit produced the remelting of the chip despite lubrication (not shown). Further work is required to optimize spacing, multiple passes and tool load. Metal, however, is preferable because of the better cleaning properties of the metal mold and because metal can be recycled versus disposable plastic, in the interest of process sustainability.

The macroscopic bending of the adjacent groove and the fact that the yield strength was overcome is visible from the permanent bending of walls and reclosing of grooves, at cutting speed as low as 0.5 m/min, in single passes, due to the tool pressure (200 MPa according to the literature). The last pass (cuts go from right to left) is not reclosed by the next pass and is considered for the FEM validation. 

Table 1 summarizes the tested conditions. The generated surfaces form an angle γ and α with respect to the vertical and horizontal surfaces of the workpiece, respectively. With respect to orthogonal cutting, the direction of cutting has an inclination λ’, which produces a tool rake angle γ’ and a flank relief angle α’. Similarly, it can be observed from experiments that a generated groove has an internal angle β’, resulting from a cutting angle β > β’ (due to the spring back effect), which is further reduced to β’’ < β. The resulting groove angles α and γ result from the spring back effect of the tool flank angle α’ and rake angle γ’, defined along a cutting trajectory with direction λ’.

## 3. FE Model for the Incipient Chip Formation Process

The chip formation problem was modeled by different commercially available FE solutions, which are critically compared.

### 3.1. LS-Dyna FE Model

Ansys LS-Dyna V4.7.17, Canonsburg, PA, USA, commercially available software was used to simulate incipient cutting process of the materials tested. Boundary conditions were defined. The created tool and its dimensional properties used in the simulation were determined considering the tool used in the experimental work. The cutting tool is considered to be rigid and meshed, containing 2700 elements. The cutting tool used was coated steel and it did not have any groove on the rake face. The values shown in Table 2 are used for the mechanical properties of the cutting tool.

The workpiece is modeled and meshed by 112,500 Smooth Particles Hydrodynamics (SPH) elements. Mesh, dimension, and boundary conditions are given in Figure 5. The SPH meshing method was used instead of Element Free Galerkin (EFG) adaptive mesh method [20] because it is faster to model and calculate; moreover, in this preliminary exploration, no quantitative analyses have been carried out. In the application of the SPH model, the Johnson–Cook material model with accumulation of damage, the coefficient of distortion, the equation of state and additional thermal parameters such as thermal softening exponent, hardening exponent, melting temperature and specific heat were used for metallic materials. The value of these parameters is presented in Table 3. During the cutting process, the heating of the workpiece is considered, and the contact type has been selected regarding thermal parameters of process interaction.

The material model used for aluminum and steel was Johnson–Cook. Delrin POM-C was modeled using piecewise linear plasticity. Table 3 presents the mechanical properties of aluminum, steel, and polymer. Al 5083 H116 was considered in the FEM of aluminum alloy compared with machined Al 6060-T6, and rescaled to conform the depth of cut d. The phenomenological chip formation and quantitative deformation are consistent. In addition, these two aluminum alloys have close material and thermomechanical properties, with regard to comparing the generated shapes in simulations and experimental tests.

### 3.2. Abaqus FE Model

The FE model consists of a 1000 × 300 μm 2D workpiece [21,22,23]. The tool is considered perfectly rigid and presents a rounded tip fillet whose radius is 0.75 μm. The workpiece is discretized into about 20,000 CPE4RT elements (plane strain linear elements with four nodes per element, reduced integration and thermal effect), for which the element deletion flag has been activated, with a maximal degradation factor of 0.99. The mesh has a resolution of 2.5 μm and can be seen in Figure 6. The adaptive meshing algorithm Arbitrary Lagrangian Eulerian (ALE), available in Abaqus, was used to mitigate the effect of the severe mesh deformation during the simulations. The cutting tool is meshed by 100 CPE4RT elements (the tool is, however, rigid). The interaction between the tool and the workpiece considers both the tangential friction effect (the friction coefficient is equal to 0.2) and the normal non-penetration condition (hard contact). The overall geometry and boundary conditions are reported in Figure 5. For the tip of the tool, prescribed horizontal and vertical displacement components are applied. The solution is performed via Abaqus/Explicit solver. When elements achieve the degradation level, they are deleted from the mesh, so that the cut appears. It is important to note that the cut is not a “zero volume” one since the degraded elements are deleted during the simulation.

## 4. Material Modeling

To model the plastic behavior of aluminum and steel, the Johnson–Cook constitutive equation is employed, which can be represented in equation (4):(4)$$\sigma_{y}=\left(A+B{\underline{\varepsilon }}^{p^{n}}\right)\left(1+c\;lnln\;{\dot{\varepsilon }}^{*}\;\right)\left(1-T^{{*}^{m}}\right)$$
where *A*, *B*, *C*, *n*, and *m* are user defined input constants, and ${\underline{\varepsilon }}^{p}$ is the effective plain strain,
(5)$${\dot{\varepsilon }}^{*}=\frac{\underline{{\dot{\varepsilon }}^{p}}}{{\dot{\varepsilon }}_{0}}$$
is the effective plastic strain rate for:(6)$${\dot{\varepsilon }}_{0}=1\;s^{-1}$$
(7)$$T^{*}=\frac{T-T_{Room}}{T_{Melt}-T_{Room}}$$

This material type is applicable to the high-rate deformation of many materials, including most metals. The Johnson–Cook model remains valid down to lower strain rates and even into the quasi-static regime. Typical applications include explosive metal forming, ballistic penetration, and impact.

In the Johnson–Cook constitutive equation, σ expresses the flow stress. The workpiece temperature is expressed as T; the workpiece melting temperature is T_m_, and the room (reference) temperature T_0_ is 20 °C. n is the hardening coefficient; m is the thermal softening coefficient. Coefficient A is the yield strength, B is the hardening modulus, and C is the strain rate sensitivity coefficient [17].

The model behavior of Delrin material piecewise linear plasticity is presented in Equations (8) and (9):(8)$$\emptyset =\frac{1}{2}s_{ij}^{2}-\frac{\sigma_{y}^{2}}{3}\leq 0$$
(9)$$\sigma_{y}=\beta \left[\sigma_{0}+f_{h}\left(\varepsilon_{eff}^{p}\right)\right]$$
here the hardening function $\varepsilon_{eff}^{p}$ can be specified in tabular form as an option.

The elastic part is defined by the Young’s modulus and Poisson ratio. The plasticity can be defined with different approaches. We use multi-plasticity curves (stress vs. plastic strain) for the strain rate range. The failure and damage model have not been considered. The comparison between simulation and experiment shows that this model gives the correct direction for the next development.

The Johnson–Cook and piecewise linear plasticity parameters are given in Table 3. Al 5083 H116 has been used as the closest aluminum to the one used in experiments already characterized in LS-Dyna. For ongoing systematic quantitative comparison analyses, software tools to embed new materials are available.

## 5. Results

### 5.1. LS-Dyna Results

The generated chips and shape resulting from the micro incipient cutting process of aluminum, steel and plastic workpiece are presented in Figure 4a,b. The FE model by LS-Dyna is capable to capture the experimental results, as shown in Figure 7a–c. The obtained and predicted incipient chip shapes for different materials are consistent.

The two different material models, respectively, Johnson–Cook and piecewise linear plasticity, used for FE modeling of the incipient cutting process of aluminum/steel and plastic, were shown as good choices.

The SPH meshing method was compatible and sufficient for modeling.

The processing time between 2 and 4 h on 64-bit 2 cores Intel^®^ i-7-5600U processor @ 2.6 GHz 8 GB RAM 11 GB virtual memory seems acceptable for extensive simulation plans.

This method is especially useful to simulate the effect of subsequent passes, as shown in Figure 4b, where cuts are reclosed.

In all the cases represented in Figure 7a–c, a good quality of the bottom of the cut was achieved, which is a major requirement for the intended target application. Currently, an ideal null radius for the tool is used; the tool tip is subject to quick wear with harder materials such as aluminum and steel vs. plastic. The FEM simulation can indirectly allow an estimate of the tool duration by predicting how the actual tool tip radius affects the groove bottom radius.

The FEM model also provides the force distribution on the tool in order to prevent damage of the thin cutter in extreme conditions.

In the application of the SPH model, the Johnson–Cook material model was used with an accumulation of damage, the coefficient of distortion, the equation of state, and the additional thermal parameters. During the cutting process, the heating of the workpiece was considered. The contact type was selected regarding thermal parameters of process interaction.

### 5.2. Abaqus Results

Three cases were simulated: five consequent cuts considering Al 5083 H116 material with the Johnson–Cook model for plasticity and damage. Three nominal spacings between cuts were considered: 72, 100 and 120 μm. The nominal depth of cut *d* is 100 μm, whilst the flank angle α’ is 5°.

Figure 8 shows the deformation and the equivalent plastic strain (PEEQ) for a single groove. Similarly, to Section 5.1, the deformed shape and the triangular plastic region show a very good agreement with Figure 4a.

Figure 8a–c show the PEEQ for three different spacing values. The increasing reclosing of the previous cuts increases by reducing the spacing between cuts.

Figure 8d shows an example temperature distribution in the workpiece after one cut for aluminum. It can clearly be observed that the maximum temperature is under 220 °C. Consequently, the thermal effect is not relevant for the incipient cutting phenomenon but is included by the J–C simulations for completeness.

As a consequence of the degraded elements deletion, the cut surfaces have an irregular micro pattern (sawtooth), observed in all the cases and visible in Figure 8.

### 5.3. Ansys Results

A planar (2D) model of the process was also investigated in Ansys APDL, based on element 183 and shown in Figure 9a. This is probably the simplest of the tested models to achieve a preliminary investigation of different conditions with very low computational time. The material model used in this case was bilinear plasticity. A precut was considered as the initial condition, as shown in Figure 9a,b, because cutting cannot be modelled in the classic Ansys workbench vs. the LS Dyna solver available in Ansys. The configuration script is available in the Appendix A. This hypothesis provides a lower bound for the cutting forces. The simulations have shown that this simple configuration provides an acceptable solution with only few elements and allows a fast preliminary exploration of some of the cutting parameters with minimal computational resources. The parameters that can be easily tested are the rake angle, the tool inclination, the cutting depth, and the material properties of the tool and of the workpiece.

The contact surfaces are meshed with contact and target elements without friction. The simulation is carried out in two load steps, with nonlinear geometry activated to be able to simulate the large plastic deformation of the base material.

The element size is 20 µm everywhere and 10 µm on the contact surfaces.

Figure 10 shows the displacement components of nodes for aluminum (a,b) and steel (c,d) for quantitative comparison with the most complete Abaqus model and with the simple (analytical) beam model proposed below.

### 5.4. Multiple Passes in Abaqus

Figure 11a compare the angles between the groove surfaces of a single and after multiple passes for aluminum and plastic. The angle after the first cut is 28° for aluminum and 22.8°for plastic for the same cutting parameters. Most probably, this difference is due to the different elastic properties of the workpiece material. After the next passes, the angle of the previous grooves generated is reduced by 7.5 to 21.5° in aluminum and for plastic by 4.3 to 18.5°. While on one side, a lower elastic modulus produces a smaller feature (lower final internal angle β’ due to the spring back effect), a smaller interference is produced between passes. A smaller interference allows lower spacing and higher density of the desired features, yielding a better adhesion of the produced gecko material. The effect of spacing is detailed in Figure 11b.

## 6. Analytical Model for the Incipient Chip Formation

Based on the FEM simulations, a simple analytical model is proposed.

We consider a linearly elastic beam, subject to a parabolic distributed load, lying on a linear elastic foundation, as shown in Figure 12. In the solid mechanics literature, this model is referenced as Winkler’s foundation (see, for instance, [24]). The beam is hinged on one end and unconstrained to the other. The beam axis has flexural rigidity *EJ*, where *E* is the Young’s modulus and *J* is the cross sectional moment of inertia of the beam. The correlation with the actual situation is that the beam deformation describes the incipient chip formed. The elastic foundation, with stiffness constant *k*, mimics the elasticity of the bulk workpiece; the distributed load *q* models the tool pressure during cutting.

The depth of cut *d* is given. The tool slides ideally along a trajectory with inclination
λ′ = λ − β/2 − α′(10)

(See Table 3 and Figure 12) and no perturbation occurs because of the tool and fixture stiffness and of the small cut depth. 

The shape of the groove is completely defined by the deformed beam axis. We adopt an abscissa s along the undeformed beam axis, running from the hinged end to the unconstrained one. The beam length can be simply calculated as
(11)$$L=d\sqrt{1+{\left(\frac{1}{tan{\left(\lambda -\beta /2-\;\alpha \prime \right)}^{}}\right)}^{2}}$$

According to [21], the equilibrium of the loaded beam of Figure 11 is ruled by the following differential equation:(12)$$EJv^{\prime \prime \prime \prime }\left(s\right)+kv\left(s\right)=qs^{2}$$
where (.)′ denotes the derivative with respect to *s,* and *v(s)* is the displacement field of the beam axis in a direction perpendicular to the undeformed axis.

Equation (12) rewrites more compactly as
(13)$$v^{\prime \prime \prime \prime \left(s\right)}+4\Lambda^{4}v\left(s\right)=Qs^{2}$$
having defined:(14)$$\Lambda^{4}:=k/\left(4EJ\right)$$
and
(15)Q:=q/EJ

The general solution of the fourth order ordinary differential Equation (13) is:(16)$$\begin{array}{cc} v\left(s\right)=Qs^{2}/\left(4\Lambda^{2}\right) & \\ & +\sin\left(\Lambda s\right)[A\sinh\left(\Lambda s\right)\\ & +B\cosh\left(\Lambda s\right)]\\ & +\cos\left(\Lambda s\right)[C\sinh\left(\Lambda s\right)\\ & +D\cosh\left(\Lambda s\right)] \end{array}$$
where A, B, C and D are integration constants to be determined by considering the following four boundary conditions:(17)$$\begin{array}{c} v\left(0\right)=0\\ -EJv\prime \prime \left(0\right)=0\\ -EJv\prime \prime \left(L\right)=0\\ -EJv\prime \prime \prime \left(L\right)=0 \end{array}$$

This solution is valid in the elastic regime, under the action of the tool at the maximum depth. We essentially extend the elastic regime where plastic deformation would occur and suppose that the higher plastic deformation will compensate the spring back effect when the tool exits, as in the Abaqus and LS Dyna simulations.

Namely, the former two conditions of Equation (17) say that at s=0 there is a hinge (zero displacement and zero moment reaction), and the latter two conditions of Equation (17) say that at s=L there is a free end (zero shear and moment reactions).

We further consider a fifth boundary condition:(18)$$v\prime \left(0\right)=\beta +\alpha \prime$$

Equation (18) imposes the simplifying (yet reasonable) assumption that the opening angle at the groove corner point must be equal to the tool angle β plus the relief angle α′.

Conditions (17) and (18) permit to univocally determine Equation (16), by univocally expressing A, B, C, D and by characterizing Q in terms of workpiece properties and cutting parameters. 

The solution reads: (19)$$A=\left(\beta +\alpha \prime \right)\frac{sin\left(2\Lambda L\right)-sinh\left(2\Lambda L\right)}{2\Lambda {\left(cos\left(\Lambda L\right)-cosh\left(\Lambda L\right)\right)}^{2}}$$
(20)$$B=-\left(\beta +\alpha \prime \right)\frac{-cosh{\left(\Lambda L\right)}^{2}+cos\left(\Lambda L\right)cosh\left(\Lambda L\right)+sin\left(\Lambda L\right)sinh\left(\Lambda L\right)}{\Lambda {\left(cos\left(\Lambda L\right)-cosh\left(\Lambda L\right)\right)}^{2}}$$
(21)$$C=\left(\beta +\alpha \prime \right)\frac{\left(sin\left(\Lambda L\right)sinh\left(\Lambda L\right)-cos\left(\Lambda L\right)cosh\left(\Lambda L\right)+cos{\left(\Lambda L\right)}^{2}\right)}{\Lambda {\left(cos\left(\Lambda L\right)-cosh\left(\Lambda L\right)\right)}^{2}}$$
(22)D=0
(23)$$Q=-2\Lambda^{5}\left(\beta +\alpha \prime \right)\frac{\left(sin\left(2\Lambda L\right)-sinh\left(2\Lambda L\right)\right)}{{\left(cos\left(\Lambda L\right)-cosh\left(\Lambda L\right)\right)}^{2}}$$

Remarkably, the deformation and the cutting force amount depends solely on the parameter Λ, which encloses the material properties of the actual workpiece. 

As a validation of the model, we considered three sets of cutting conditions for the same material (Al 5083 H116), as reported in Figure 13. 

The model is quite close to the Abaqus solution. The analytical solution is also able to capture the curvature profile of the groove (especially in Figure 13a). It can be noticed that the analytical model is less accurate in following the groove shape for large values of s, probably because the FEM captures the dynamic of the tool during entry and exit, with local plasticity effects considered, whilst the analytical model only considers the tool while fully penetrating the workpiece in the elastic regime.

It should also be noticed that the plastic deformation predicted in Abaqus and Ansys (with precut) after the tool exit are qualitatively similar to elastic deformation predicted by the analytical model after the tool penetration.

An observation must be made regarding the fifth boundary condition. Equation (12) with the identified expressions of A, B, C, D and Q is largely independent from the net value of Λ. In fact, almost the same previsions as in Figure 13 were discovered for $\Lambda \in \left[0.003,9.5\right]$ (for the plots of Figure 12, Λ=9 was used). This suggests that the incipient chip shape slightly depends on the workpiece material, for given cutting conditions.

However, the net value of Λ deeply affects the value of the necessary force to perform the cut ($Q\propto \Lambda^{5}$). This means that the incipient chip shape will be nearly the same for different materials (for a given set of cutting parameters), but of course the needed cutting force will be different from material to material.

This fact is described by the fifth boundary condition. The cutting force depends on the cutting conditions and on the workpiece material q, and consequently, Q can be interpreted as the parabolic distributed load intensity needed to create a groove in the bulk material with an opening angle β+α′.

The right choice of Λ for different materials opens new areas of experimental and numerical investigation in the interest of more efficient incipient cutting. In fact, this parameter shall enclose the elastic as well as plastic properties of the actual workpiece. Figure 14 compares the groove profile of aluminum in Abaqus, Ansys and in the analytical model for λ = 50°, β = 30° and α’ = 5° (2D simulations). Most of the groove profile shows a very good agreement among the three methods. Particularly, the analytical and the Ansys FE model exhibit a similar profile; in the extreme regions, they diverge from the Abaqus model, which is able to catch the complex plasticity effects occurring in the incipient chip formation.

## 7. Discussion

In this first analysis of the incipient chip formation by FEMs, we explored various commercial packages to address future research. To help one to understand the differences between the tested FE models, a summary is reported in Table 4.

The thermal properties of steel and aluminum are considered by the Johnson–Cook material model, which has been used in LS Dyna and Abaqus; however, the thermal effects are clearly not relevant. 

Quantitative analyses are not the focus of this preliminary exploration, so in LS Dyna, the SPH meshing method was used instead of the EFG adaptive mesh method because it is much faster to model and calculate. Adaptive mesh was considered in Abaqus. 

LS-Dyna and Abaqus provide more accurate experimental representation and offer a wider range of simulation parameters, require higher computation time and demanding processing power. FEM parameters require references and data about thermophysical and mechanical properties, which may not be available for all the materials. Ansys is a widespread product and provides a quick preliminary exploration of parameters.

Simulations offer a map of the deformed and hardened areas, which would be very difficult to predict or measure experimentally and can provide an interference-free distance between cuts.

From Figure 15, it can be noticed that the tool penetration in plastic produces the minimum deformation surrounding the cutting area. Steel exhibits a higher presence of areas under tensile stress. The increasing compression load starting from the tool tip, which can be observed in all the cases, was captured by the analytical model.

Figure 16 shows the force increasing with penetration and the maximum value obtained in LS-Dyna for a cutting width of 1 mm.

This work has proven the shape adherence of the simulations to the desired challenging geometrical requirements. Similarly, the cutting forces are of the same order between the different FE models and the literature [10]. Despite its simplicity, the proposed analytical model has a very good agreement with the simulations; moreover, it seems to catch the main phenomena, in particular:the chip shape is mainly affected by the penetrating cutting tool angles;the sharp tip at the bottom of the groove is preserved;the load intensity depends on the workpiece material properties;the deformation of the bottom surface has been neglected, for the presence of positive relief angle, which minimizes the contact surface to the tool tip (the blade cutting edge).

This analysis can be applied to determine the optimal tool trajectory in order to minimize forces and tool wear/damage for more energy efficient cutting.

## 8. Conclusions

After many decades of the FEM simulation of cutting, this preliminary experimental exploration has shown the potential of the FEM by Ansys, LS-Dyna and Abaqus in the early transient phase of micromachining, right after the tool enters the material with a depth of less than 100 µm. An easy-to-machine metal (Al 6060-T6) and a widespread high-performance polymer (Delrin MOP-C)—each with some advantages and disadvantages as a mold material for gecko-inspired adhesives—exhibited different chip formation under the tool action, as predicted in the simulations.

These results have allowed the definition of a simple analytical model, which is able to predict the incipient chip shape in closed form, based on the geometric tool parameters. The material effect requires further experimental and analytical investigation beyond the simpler hypothesis of elastic deformation considered, despite the good agreement of the described chip morphology.

These results show the suitability of the proposed cutting approach by a diamond coated thin blade, which overcomes intrinsic limitations of alternative cutting methods experimented, and is briefly presented in the paper to show the better sustainability of the alternative process proposed here. The quality of the achieved geometry has also been confirmed by cutting experiments and by comparing the three different FE models. The combination of elastic spring back and plastic permanent deformation offers a variability of cutting parameters to be simulated in order to achieve the desired final shape as a result of incremental deformations, caused by successive passes. Further FEM analyses are required to investigate this new area of incipient cutting.

The main scientific merit of the current work is the analysis and description of the material reaction in the transition from indentation to cutting in the very early phase of the tool penetration. A qualitative convergence among conceptually different approaches —theoretical, numerical, and experimental—has been shown and formally described. 

The main practical outcome is to model a new cutting process for micromachining—such as the challenging gecko features—and may be extended to texturing functional surfaces.

Additional research is required to extend the simulations to different materials and cutting conditions. By the FEM, a number of combinations can be simulated and fine-tuned to determine the optimal cutting parameters to achieve the desired microscopic features, such as the complex gecko material geometry and similar introduced in this paper, and to minimize the cutting forces and energy for more efficient cutting.

## Figures and Tables

**Figure 1 materials-14-03789-f001:**
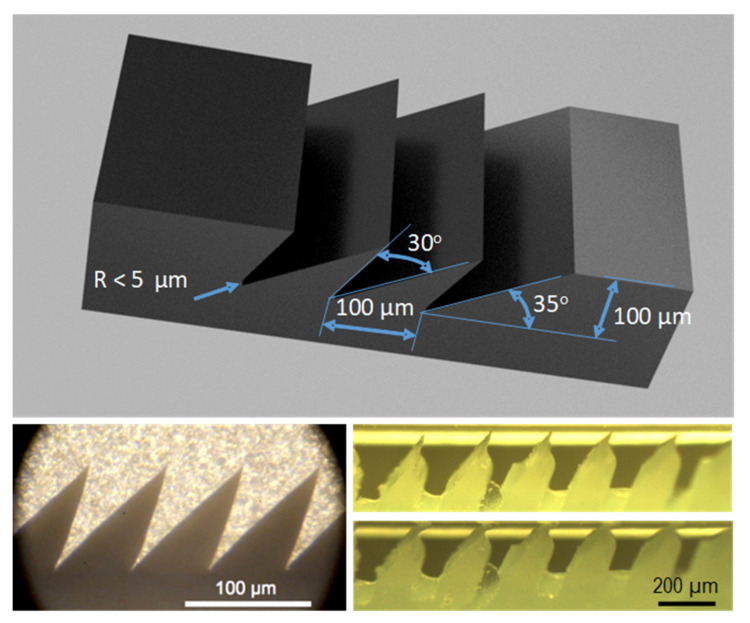
Geometrical specifications for the durable mold for gecko patches. Bottom left: micrograph of micromachined wax mold shapes (from [7] with permission from Mark Cutkosky, Stanford University). Bottom right: silicone gecko patch as cast (alternative hierarchical geometry from [8]) in unloaded and loaded conditions, showing the importance of the sharp tip for directional adhesion engagement.

**Figure 2 materials-14-03789-f002:**
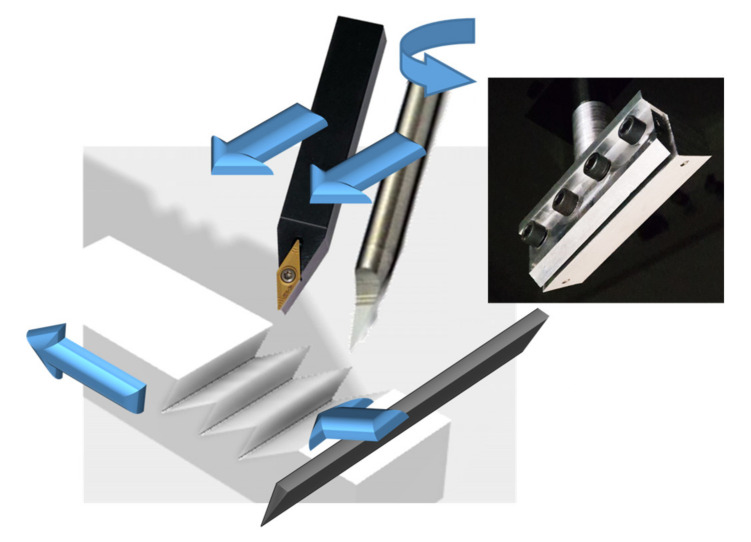
Overview of the cutting strategy to fabricate a permanent mold by a special blade and its support (inset). The alternative strategy providing the results in Figure 3 entails a chisel or a traditional single point tool along the transverse direction. A range of materials was tested using low viscosity mineral oil (WD-40) as the lubricant. Plastic and metals were tested for different objectives, which is explained in the next section, and for performance comparison of the tested solutions.

**Figure 3 materials-14-03789-f003:**
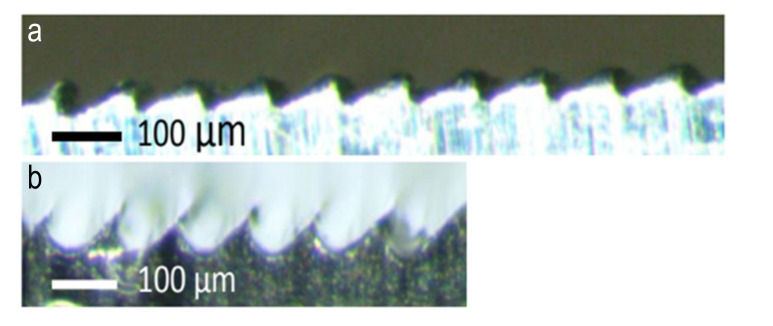
Micrograph of machined Al 7075-T6 (**a**) and AISI 4140 steel (**b**).

**Figure 4 materials-14-03789-f004:**
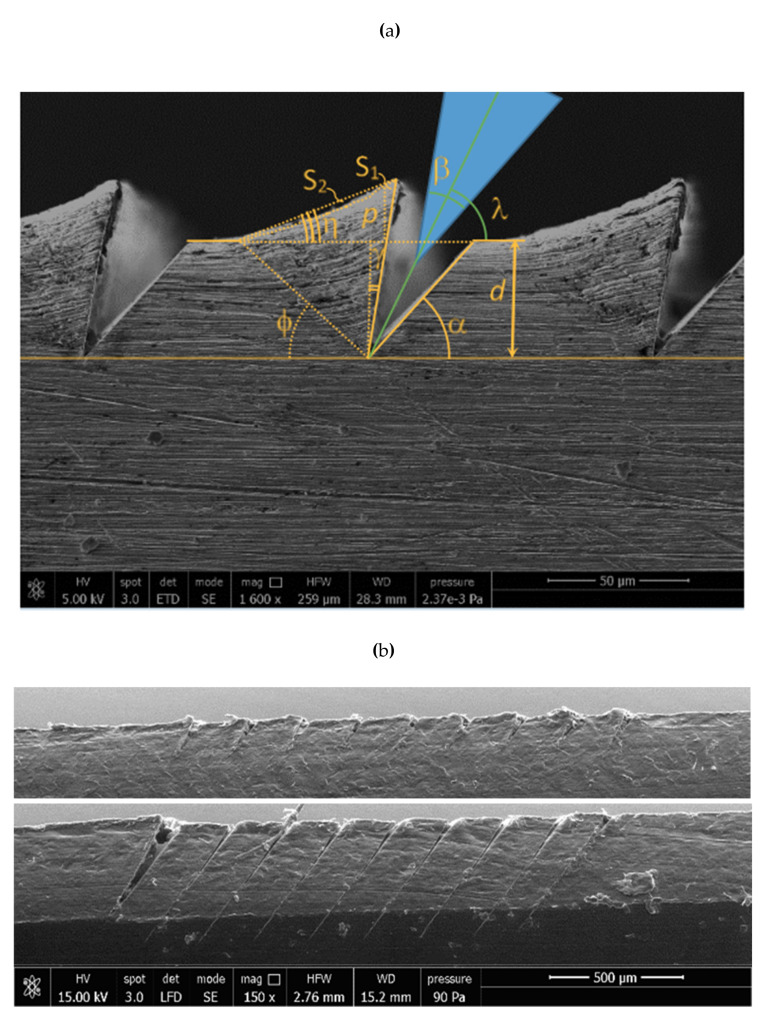
(**a**) SEM micrograph of durable mold made by Al 6060-T6: wedge angle definitions. Plastic deformation is enhanced by the deflection of stripes. (**b**) SEM micrograph of acetal (yield strength 68 MPa) with blade cutting at 100 (top) and 500 (bottom) µm depth d.

**Figure 5 materials-14-03789-f005:**
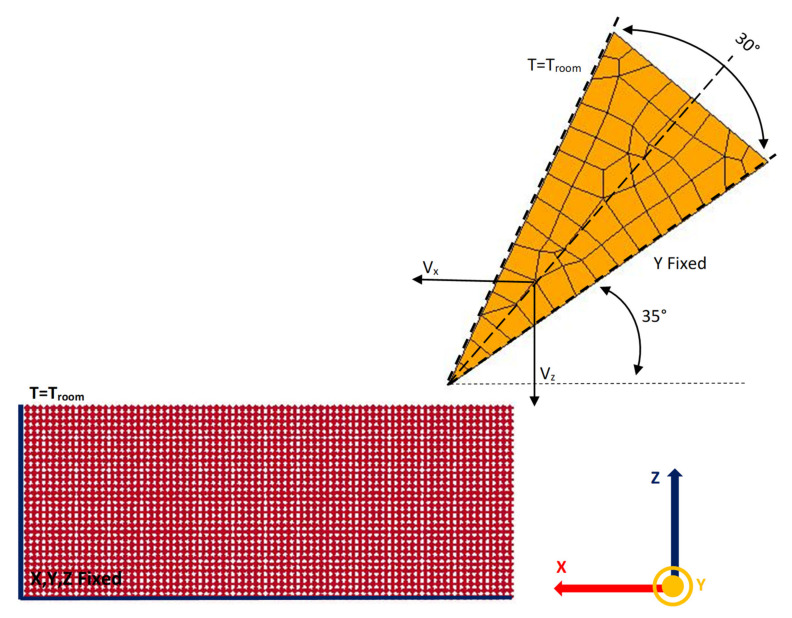
Meshing and boundary conditions for workpiece and tool in LS Dyna.

**Figure 6 materials-14-03789-f006:**
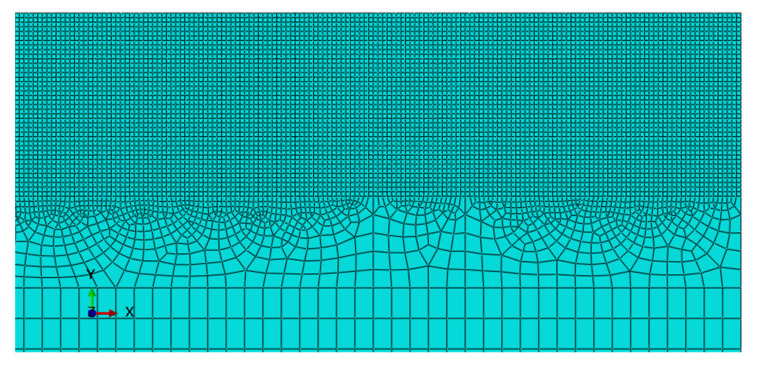
Meshing detail of the workpiece in Abaqus. The configuration script is available in the Appendix A.

**Figure 7 materials-14-03789-f007:**
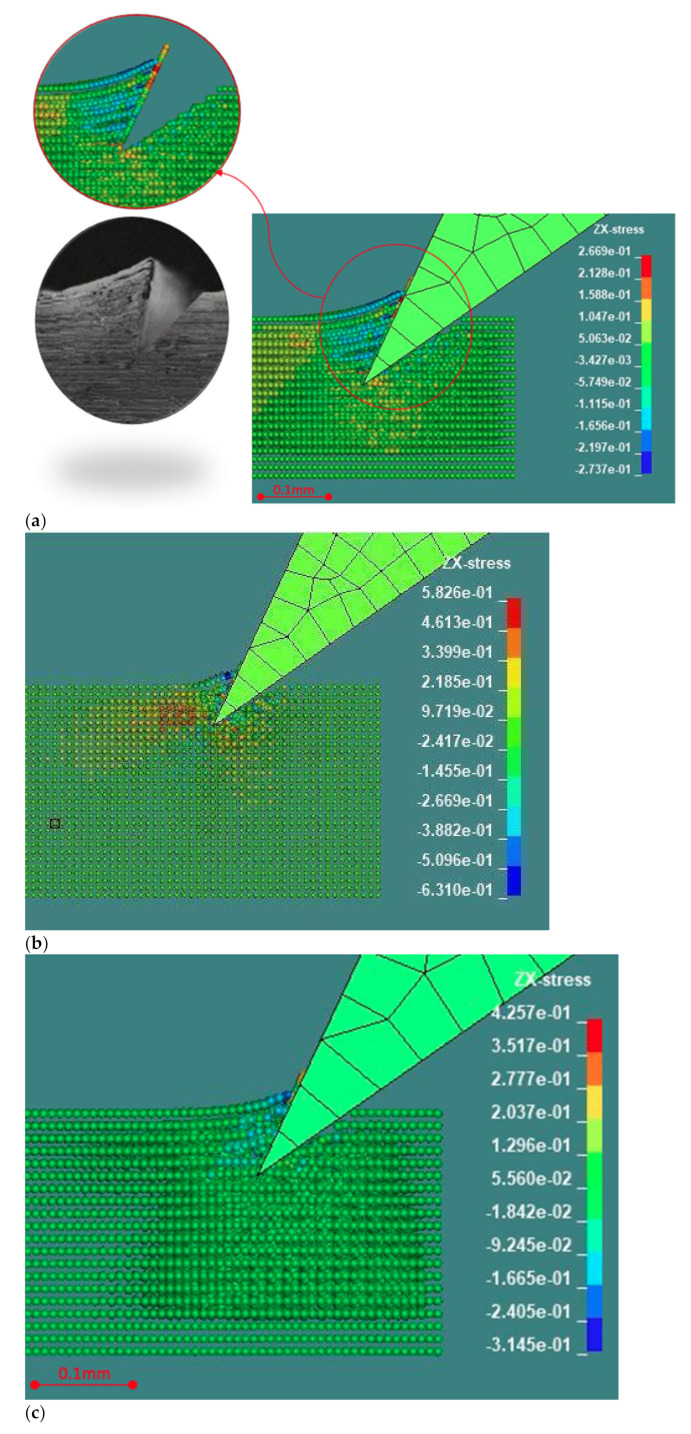
(**a**) LS-Dyna FEM simulation of Al 5083 H116 compared with machined Al 6060-T6 rescaled to conform the depths of cut (100 vs. 80 µm). The phenomenological chip formation and quantitative deformation are consistent. A 2D video simulation is available in the Appendix A. (**b**) LS-Dyna FEM simulation of AISI 1040. Improvement over alternative processes in Figure 3 is foreseen regarding the smaller bottom radius. A 3D video simulation is available in the Appendix A. (**c**) LS-Dyna FEM simulation of Delrin POM-C. The phenomenological chip formation and quantitative deformation are consistent with Figure 4b after scaling to accommodate the different depths of cut.

**Figure 8 materials-14-03789-f008:**
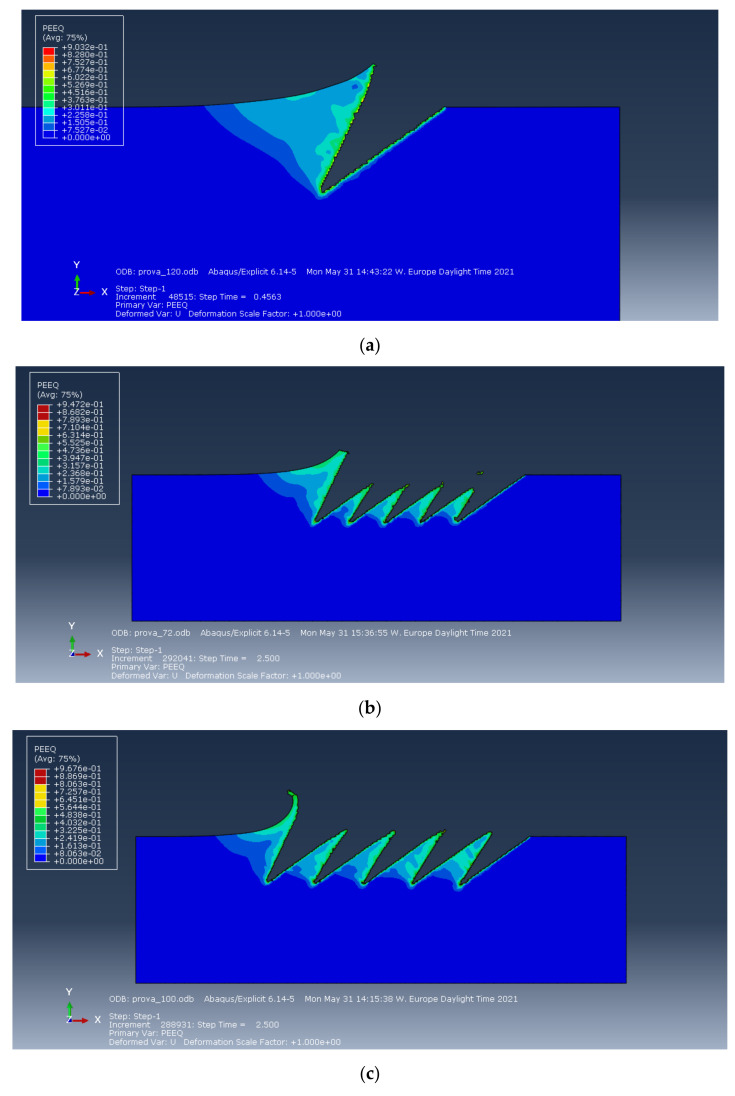
**(a)** Equivalent plastic strain for a single first cut in aluminum. (**b**) Equivalent plastic strain for spacing of 72 µm in Abaqus in aluminum. (**c**) Equivalent plastic strain for spacing of 100 µm in Abaqus in aluminum. (**d**) Equivalent plastic strain for spacing of 120 µm in Abaqus in aluminum. (**e**) Temperature after one cut (in Kelvin) in aluminum.

**Figure 9 materials-14-03789-f009:**
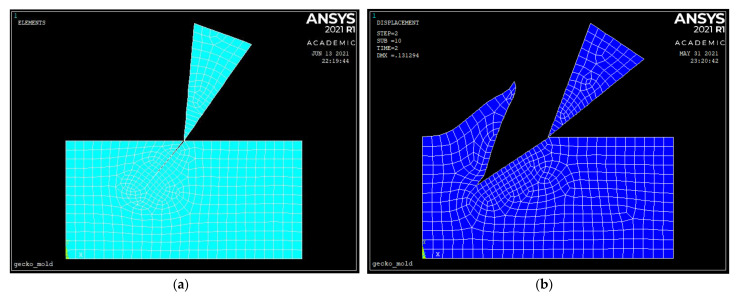
(**a**) Meshing of the workpiece and tool in Ansys, with precut along the cutting trajectory, with inclination λ’ and depth d. The right and the lower edges of the workpiece are constrained in all the directions; the displacement components are applied to the blade, which is considered rigid. (**b**) Ansys simulation of the formed chip and of the deformed mesh in aluminum with a precut on the workpiece.

**Figure 10 materials-14-03789-f010:**
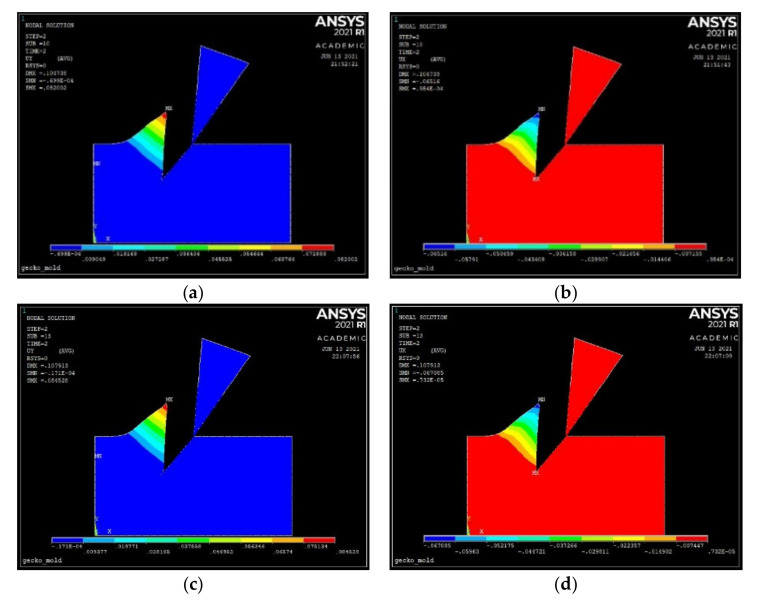
(**a**) Vertical and (**b**) horizontal displacement components for aluminum. (**c**) Vertical and (**d**) horizontal displacement components for steel.

**Figure 11 materials-14-03789-f011:**
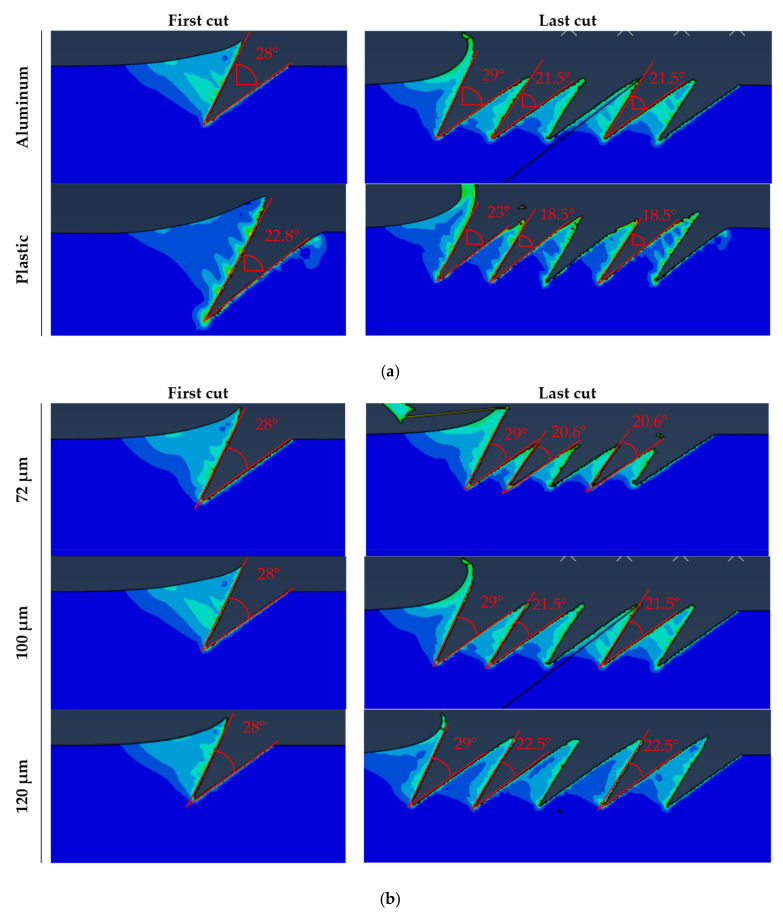
(**a**)**.** Interference between subsequent passes for two different materials simulated with Abaqus with 100 µm spacing. (**b**). Interference between subsequent passes for aluminum as a function of spacing between passes simulated with Abaqus.

**Figure 12 materials-14-03789-f012:**
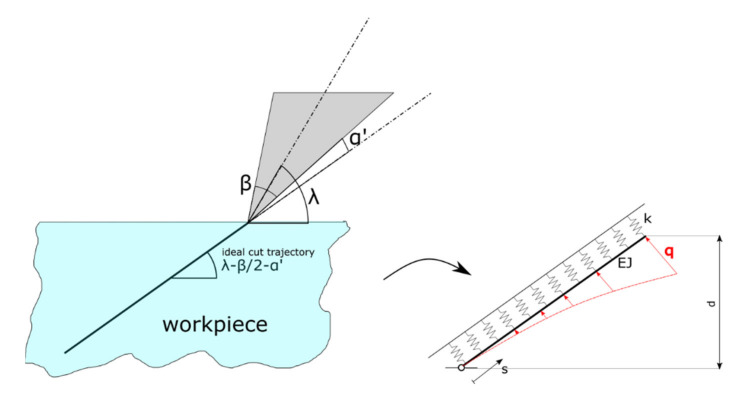
The proposed beam model for the prediction of the deformation of top surface of the groove generated during the incipient chip formation.

**Figure 13 materials-14-03789-f013:**
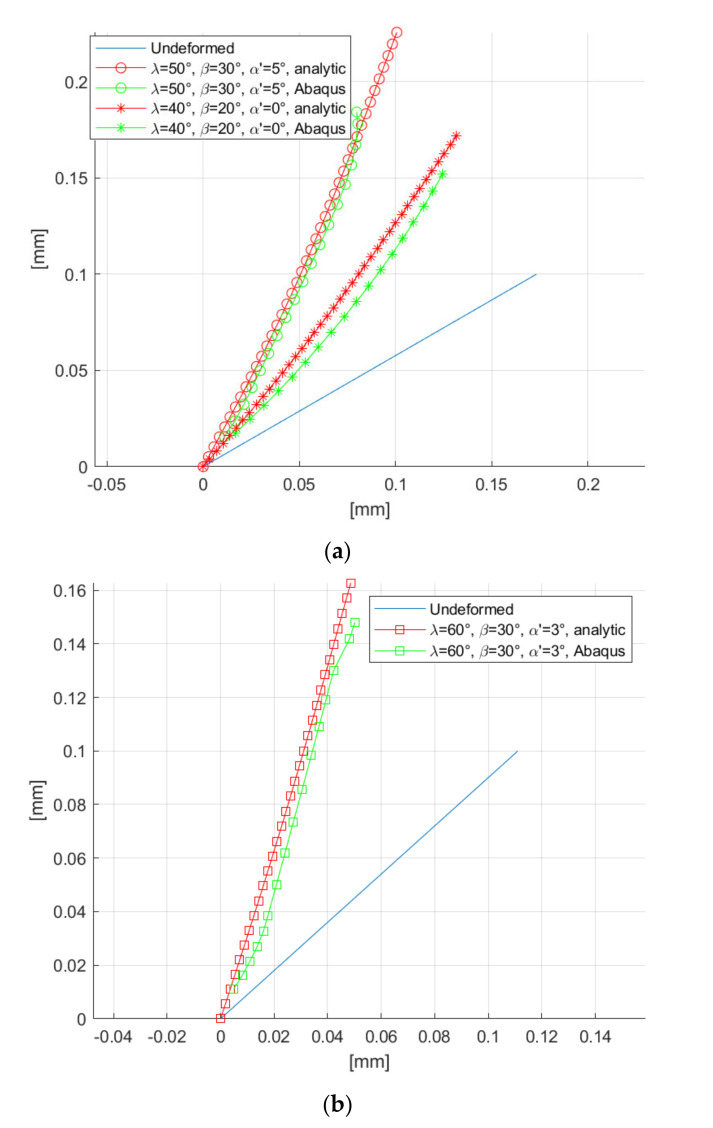
Validation of the proposed beam model for different cutting trajectories: λ′ = 30° (**a**) and λ′ = 42° (**b**).

**Figure 14 materials-14-03789-f014:**
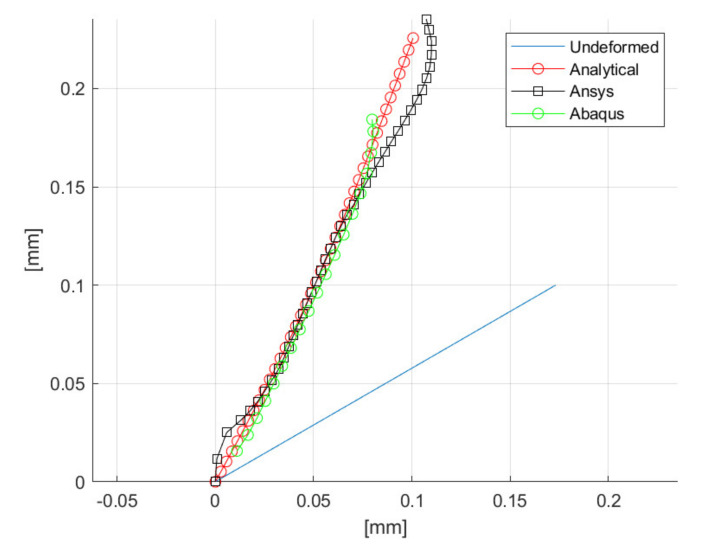
Comparison of groove profiles for a cutting tool trajectory λ′ = 30° simulated analytically, in Ansys and Abaqus.

**Figure 15 materials-14-03789-f015:**
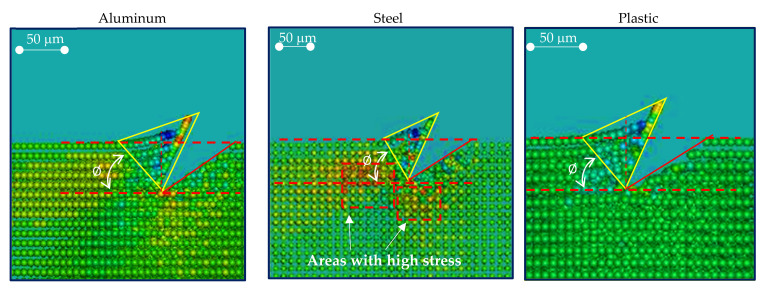
Comparison of the deformed areas for aluminum, steel, and plastic in the simulation with LS-Dyna.

**Figure 16 materials-14-03789-f016:**
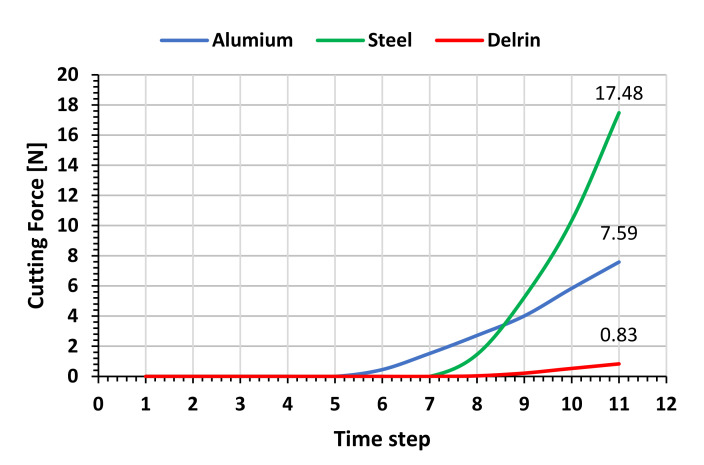
Resulting force for aluminum, steel, and plastic from simulations with LS-Dyna during the tool penetration into the workpiece.

**Table 1 materials-14-03789-t001:** Experimental and simulated conditions according to the symbols in Figure 4a.

Parameter	Symbol	Value (Range)	Units
depth of cut	*d*	50–100	µm
tool (wedge) cutting angle	β	24–34	°
tool radius	ρ	0–0.9	µm
tool inclination	λ	25–35	°
cutting trajectory	λ′	30–50	°
groove top surface angle	γ	–	°
groove bottom surface angle	α	–	°
groove internal angle (single cut)	β′	22–35	°
groove internal angle (previous pass)	β″	18–23	°
tool rake angle	γ′	0	°
tool flank angle	α′	0–5	°

**Table 2 materials-14-03789-t002:** Properties of the tool material [16].

Properties (Units)	Value
Poisson’s ratio	0.3
Density (kg/m^3^)	7850
Young’s modulus (GPa)	200

**Table 3 materials-14-03789-t003:** Johnson–Cook and piecewise linear plasticity model parameters for the aluminum, steel and polymer tested in LS-Dyna [18,19].

Parameter	Symbol (unit)	J-cook		P-L-P
		Al 5083 H116	AISI 1040	Delrin POM-C
Density	*Ro* (kg/m^3^)	2700	7810	1420
Shear modulus	*G* (Pa)	2.69 × 10^10^	78.5 × 10^10^	-
Elastic modulus	*E* (Pa)	70 × 10^9^	204 × 10^9^	4 × 10^10^
Poisson’s ratio	*PR*	0.33	0.3	0.37
Yield stress	*A* (Pa)	1.67 × 10^8^	5.53 × 10^8^	7 × 10^8^
Hardening constant	*B* (Pa)	5.96 × 10^8^	6.01 × 10^8^	-
Strain-rate constant	*C*	0.001	0.0134	-
Thermal softening exponent	*M*	0.859	1	-
Hardening exponent	*N*	0.551	0.234	-
Melting temperature	*TM* (^o^C)	670	1480	-
Room temperature	*TR* (^o^C)	20	20	-
Ref. strain rate	*EPSO* (s^−1^)	1	1	-
Specific heat	*CP* (J kg^−1^ K^-1^)	910	642	-
Pressure cutoff	*PC* (Pa)	−1.5 × 10^9^	-	-
Failure parameter	*D1*	0.0261	1.2	-
Failure parameter	*D2*	0.263	-	-
Failure parameter	*D3*	−0.349	-	-
Failure parameter	*D4*	0.247	-	-
Failure parameter	*D5*	16.8	-	-

**Table 4 materials-14-03789-t004:** FE models summary. Processing time is on 2.6 GHz 64-bit processor with 8 GB RAM PC.

	Ansys	LS-Dyna	Abaqus
2D/3D	2D	3D	2D (plane strain)
Element number	600	110,000	20,000
Average element size (µm)	20	20	5
Element type	plane 183	sphere	CPE4RT
Wedge Blade	rigid	rigid	rigid
Adaptive mesh	no	no	yes
Tangential friction coefficient	0	0.2	0.2
Thermal effect	no	yes	yes
Plasticity model	bilinear	Johnson–Cook	Johnson–Cook
Damage model	-	Johnson–Cook	Johnson–Cook
Cut initialization	yes	no	no
Processing time per groove (min.)	5	180	30

## Data Availability

The data presented in this study are available on request from the corresponding author.

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
