# Peer review of "FEM and Analytical Modeling of the Incipient Chip Formation for the Generation of Micro-Features"

_materials, 2021, doi:10.3390/ma14143789_

Round 1

Reviewer 1 Report

The article deserves special recognition. The authors performed numerical simulations of a actual physical process using independent numerical simulation programs (LS-Dyna, Abaqus, and Ansys). The paper is suitable for publication in its present form. The purpose of the paper has been defined and realized in the presented research results. For the future, however, I would suggest significantly expanding the literature references used by the authors. Nevertheless, the content of the paper does not raise any objections and can be published in its present form.

Author Response

Answers to Reviewer 1 

The article deserves special recognition. The authors performed numerical simulations of a actual physical process using independent numerical simulation programs (LS-Dyna, Abaqus, and Ansys). The paper is suitable for publication in its present form. The purpose of the paper has been defined and realized in the presented research results. 

Answer: First, we would like to thank Reviewer 1 for the review and positive comments. 

For the future, however, I would suggest significantly expanding the literature references used by the authors. Nevertheless, the content of the paper does not raise any objections and can be published in its present form.

Answer: Thank you for addressing the opportunity of expanding the literature. Nine new references (from 17 to 26) have been added to the paper. 

Reviewer 2 Report

  1. This paper shows the potential of FEM analysis by Ansys, LS-Dyna and Abaqus. in the phase of micromachining. The results showed better sustainability and suitability of the proposed cutting approach, overcoming some limitations of alternative cutting methods. The quality of achieved geometry has been confirmed. 
  2. I do not have any major weaknesses in the paper. The paper is well-organized, experiments are clearly explained with all necessary information. The topic seems interesting. The Authors compared the proposed approach with the existing ones, which is an advantage. 
  3. I do not have any major improvement to suggest. 
  4. Minor suggestions are:
  1. Fig. 1 should be improved - numbers and letters are hard to read
  2. In Table 3. - for the direction of cutting - there is possibly an extra circle after 30-50
  3. In Table 4. - the authors might consider writing the unit Kg/m^3 to kg/m^3
  4. After Eq. 4 - the word "and" should not be italic (A, B, C, n and m)
  5. Eq. 9 can be revised
  6. In Table 5. - at position Hardening constant/AISI 1040 - revise large X
  7. Fig. 7 - extra dot at the end of the caption
  8. The Authors might consider upgrading the literature with few more recent articles.

Author Response

Answers to Reviewer 2 

This paper shows the potential of FEM analysis by Ansys, LS-Dyna and Abaqus. in the phase of micromachining. The results showed better sustainability and suitability of the proposed cutting approach, overcoming some limitations of alternative cutting methods. The quality of achieved geometry has been confirmed. 

I do not have any major weaknesses in the paper. The paper is well-organized, experiments are clearly explained with all necessary information. The topic seems interesting. The Authors compared the proposed approach with the existing ones, which is an advantage. 

I do not have any major improvement to suggest. 

Answer: First, we would like to thank Reviewer 2 for the review and positive comments. 

Minor suggestions are:

Fig. 1 should be improved - numbers and letters are hard to read

In Table 3. - for the direction of cutting - there is possibly an extra circle after 30-50

In Table 4. - the authors might consider writing the unit Kg/m^3 to kg/m^3

After Eq. 4 - the word "and" should not be italic (A, B, C, n and m)

Eq. 9 can be revised

In Table 5. - at position Hardening constant/AISI 1040 - revise large X

Fig. 7 - extra dot at the end of the caption

The Authors might consider upgrading the literature with few more recent articles.

Answer: All the suggestions have been followed slavishly. Nine new references have also been added, including two since 2020. Thank you.

Reviewer 3 Report

In this manuscript, the authors mainly presented the simulations of incipient chip formation for the generation of micro features using FEM methods. Three different commercial softwares, i.e. LS-Dyna, Abaqus, Ansys were utilized, and two classes of materials, i.e. metal and polymer were investigated. The authors claimed the modeling results were consistent with experimental observation with respect to the deformation.

The authors are advised to address the following comments:

Major comments

  1. The novelty of this paper is not convincing from the perspective of the reviewer. Although the authors mentioned the lack of FEM analysis for the modeling of incipient chip formation, based on the FEM process described in this paper, the reviewer did not see the much difference from the other chip formation simulations that are widely existed in the literature. The FEM implementation of this work seems quite common. Hence, the authors are suggested to give more details about the specific contributions in this paper, either from experiment or modeling.
  2. In the FEM simulations, the authors didn’t consider the thermal effect and adaptive mesh (shown in Table 6), which are important in reality for accurate modeling. The assumption of avoiding them should be justified properly in this work.
  3. The description of the FEM simulation using Ansys in Section 5.3 should be more detailed if this part is important. For example, in Figure 9, what was the material used and loading condition? And the previously cut model as the initial condition should be also presented for better understanding of authors' purpose.

Minor comments

The minor comments related to problematic labeling, typo, etc., have been detailed in the marked manuscript in the attachment. Please address them accordingly.

Author Response

Answers to Reviewer 3 

In this manuscript, the authors mainly presented the simulations of incipient chip formation for the generation of micro features using FEM methods. Three different commercial softwares, i.e. LS-Dyna, Abaqus, Ansys were utilized, and two classes of materials, i.e. metal and polymer were investigated. The authors claimed the modeling results were consistent with experimental observation with respect to the deformation.

Answer: First, we would like to thank Reviewer 3 for the review. 

The authors are advised to address the following comments:

Major comments

The novelty of this paper is not convincing from the perspective of the reviewer. Although the authors mentioned the lack of FEM analysis for the modeling of incipient chip formation, based on the FEM process described in this paper, the reviewer did not see the much difference from the other chip formation simulations that are widely existed in the literature. The FEM implementation of this work seems quite common. Hence, the authors are suggested to give more details about the specific contributions in this paper, either from experiment or modeling.

Answer: In order to enhance the novelty of the proposed work, which was initially restricted to the numerical modeling of the incipient chip formation, a simple analytical model has also been developed and included in the revised paper. 

The main benefit of the proposed analytical model is to catch the main phenomena observed during the simulation, which have not yet been described in the literature. 

These concepts are now included in the new Section 7 of the revised paper. 

In the FEM simulations, the authors didn’t consider the thermal effect and adaptive mesh (shown in Table 6), which are important in reality for accurate modeling. The assumption of avoiding them should be justified properly in this work.

Answer: Thank you for pointing out the importance of the thermal effects. The thermal properties of steel and aluminum are considered by the Johnson-Cook material model, which is considered in LS Dyna and Abaqus.  

The new Figure 8.d shows an example temperature distribution in the workpiece after one cut for aluminum. It can be clearly observed that the maximum temperature is under 220°C. Consequently, the thermal effect is not relevant for the incipient cutting phenomenon, but is included during the simulations for completeness. 

Regarding your observation about the use of adaptive mesh, quantitative analyses were not the focus of this preliminary exploration, so in LS Dyna Smooth Particles Hydrodynamics SPH meshing method was used instead of EFG adaptive mesh method because it is way faster to model and calculate. Adaptive mesh was considered in Abaqus. 

These important assumptions have been recalled at the outset of the revised discussion and have been specified in Table 6. 

The description of the FEM simulation using Ansys in Section 5.3 should be more detailed if this part is important. For example, in Figure 9, what was the material used and loading condition? And the previously cut model as the initial condition should be also presented for better understanding of authors' purpose.

Answer: More explanations about the Ansys model have been added to the revised section 5.3, along with additional simulations. 

The precut model as the initial condition has now been explicitly introduced.

Minor comments

The minor comments related to problematic labeling, typo, etc., have been detailed in the marked manuscript in the attachment. Please address them accordingly.

Answer: All the comments in the provided attachment have been followed slavishly. Thank you.

Round 2

Reviewer 3 Report

Dear authors,

Thanks for your efforts on revising the manuscript accordingly.

Overall, the reviewer is satisfied with the updated version which addressed the major comments by providing more details and additional materials.

It is observed the scientific soundness and quality of presentation have been much improved.

A few minor comments are given as follows:

  1. Although the presented results of using the analytic model look fine, the reviewer is somewhat skeptical of the applicability of adopting such elastic analytical model to analyze the process that apparently involves the plasticity(Figure 8), it is suggested to mention the limitation when the analytic model is introduced or applied.
  2. The format of listed references are not consistent, please check and refer to the standard of journal.

Otherwise, the paper is well written and recommended for publication.

Author Response

Answers to Reviewer 3 

Dear authors,

Thanks for your efforts on revising the manuscript accordingly.

Overall, the reviewer is satisfied with the updated version which addressed the major comments by providing more details and additional materials.

It is observed the scientific soundness and quality of presentation have been much improved.

Answer: First, we would like to thank Reviewer 3 also for this review and appreciation. 

A few minor comments are given as follows:

  • Although the presented results of using the analytic model look fine, the reviewer is somewhat skeptical of the applicability of adopting such elastic analytical model to analyze the process that apparently involves the plasticity(Figure 8), it is suggested to mention the limitation when the analytic model is introduced or applied.
  • The format of listed references are not consistent, please check and refer to the standard of journal.

Otherwise, the paper is well written and recommended for publication.

Answer: The limitation of the elastic analytical model is now clearly specified also when it is introduced and applied, pointing out its motivations and limitations. 

Following this same approach, we have taken the opportunity of this revision to enhance the main contributions of this paper. 

All the references have been reformatted according to the journal directions.

Thank you.